# A Novel Source/Drain Extension Scheme with Laser-Spike Annealing for Nanosheet Field-Effect Transistors in 3D ICs

**DOI:** 10.3390/nano13050868

**Published:** 2023-02-26

**Authors:** Sanguk Lee, Jinsu Jeong, Bohyeon Kang, Seunghwan Lee, Junjong Lee, Jaewan Lim, Hyeonjun Hwang, Sungmin Ahn, Rockhyun Baek

**Affiliations:** Department of Electrical Engineering, Pohang University of Science and Technology (POSTECH), Pohang 37673, Gyeongbuk, Republic of Korea

**Keywords:** nanosheet FET, laser-spike annealing, rapid thermal annealing, lateral diffusion, strain engineering, 3D ICs, junction technology, technology-computer-aided-design simulation

## Abstract

This study proposed a novel source/drain (S/D) extension scheme to increase the stress in nanosheet (NS) field-effect transistors (NSFETs) and investigated the scheme by using technology-computer-aided-design simulations. In three-dimensional integrated circuits, transistors in the bottom tier were exposed to subsequent processes; therefore, selective annealing, such as laser-spike annealing (LSA), should be applied. However, the application of the LSA process to NSFETs significantly decreased the on-state current (I_on_) owing to diffusionless S/D dopants. Furthermore, the barrier height below the inner spacer was not lowered even under on-state bias conditions because ultra-shallow junctions between the NS and S/D were formed far from the gate metal. However, the proposed S/D extension scheme overcame these I_on_ reduction issues by adding an NS-channel-etching process before S/D formation. A larger S/D volume induced a larger stress in the NS channels; thus, the stress was boosted by over 25%. Additionally, an increase in carrier concentrations in the NS channels improved I_on_. Therefore, I_on_ increased by approximately 21.7% (37.4%) in NFETs (PFETs) compared with NSFETs without the proposed scheme. Additionally, the RC delay was improved by 2.03% (9.27%) in NFETs (PFETs) compared with NSFETs using rapid thermal annealing. Therefore, the S/D extension scheme overcame the I_on_ reduction issues encountered in LSA and significantly enhanced the AC/DC performance.

## 1. Introduction

Advanced semiconductor technologies have driven scaling and performance enhancement from planar transistors to fin-shaped field-effect transistors (FinFETs) [1,2,3,4,5,6]. So far, in FinFETs, a larger effective channel width can be achieved by increasing the fin height, and the gate controllability over the channel has been improved by scaling the fin width [4,5,6]. However, the scaling of the fin-shaped channel has some shortcomings, as the fin formation process involves nonuniform fin width and line edge roughness variations [7,8,9]. Conversely, the nanosheet (NS) channels of nanosheet FETs (NSFETs) are less sensitive to process variations because they are formed by epitaxial growth [10,11]. Additionally, because the gate metal surrounds the NS channels, the gate controllability over the channel is enhanced, and improved gate controllability effectively suppresses the off-state current (I_off_). Moreover, because the channel width in the same footprint can be take a wider form than that in FinFETs, a higher on-state current (I_on_) can be achieved [10,12]. As a result, NSFETs have a higher I_on_/I_off_ ratio than FinFETs and are the most suitable device structure for sub-3 nm nodes and beyond.

Recently, three-dimensional integrated circuits (3D ICs) have been extensively investigated to increase the number of transistors in a given area [13,14]. Three-dimensional ICs constitute one of the key technologies that can realize the highest degree of integration among currently developed technologies because transistors can be integrated on the top and bottom layers. In 3D ICs, the thermal budget during top-tier device fabrication is crucial because the thermal process in the top-tier device influences the bottom-tier device [15,16]. Therefore, the thermal budget of each process should be managed such that subsequent processes do not damage bottom-tier devices [16]. Consequently, 3D ICs require low-temperature or selective heating processes to prevent performance degradation in bottom-tier devices [17]. However, conventional source/drain (S/D) activation processes, such as rapid thermal annealing (RTA), heat the entire wafer. Therefore, the performance of bottom-tier devices inevitably deteriorates at high temperatures, resulting in performance degradation [17,18].

Because laser-spike annealing (LSA) activates S/D dopants within local and selective areas in a short time, LSA has been extensively used to moderate thermal issues. In addition, LSA using multiple beams, which can control the depth of the activation area by using different wavelengths, is also being actively studied to anneal the confined area requiring activation. Therefore, LSA can be a promising candidate for resolving performance and reliability issues encountered in 3D ICs [17,18,19].

Additionally, a small number of S/D dopants that deeply diffuse during thermal processing can cause significant performance degradation in advanced nodes, including 3D ICs [20]. For example, deeply diffused S/D dopants increase variability, leading to performance variations and a reduction in yields. Therefore, minimizing the S/D dopant diffusion is crucial to reduce variations and degradations in device behaviors. Currently, LSA is considered to be able to replace RTA in advanced nodes because it can suppress S/D dopant diffusion into the channels thanks to its short dwell time [21]. Accordingly, LSA forms an ultra-shallow junction near the interface of the channel to the S/D; thus, it is expected to suppress short-channel effects (SCEs) and performance degradation. In addition, parasitic capacitance can be significantly decreased because the overlap capacitance between the gate metal and S/D is reduced. Hence, LSA enables various technologies that are not feasible with previous technology nodes, owing to S/D dopant diffusion. Furthermore, diffusionless S/D dopants enable various device structures, which can improve AC/DC characteristics.

This study proposed an S/D extension scheme that can considerably improve the AC/DC characteristics of NSFETs by utilizing the diffusionless S/D activation of LSA for the first time. The S/D extension scheme significantly enhanced the diffusionless advantage of LSA. As a result, higher stress is induced in the NS channels so that carrier mobility can be improved. In addition, carrier concentrations in the NS channels are increased largely because the additional NS etching process extends S/D under the inner spacer regions. The effects of the proposed S/D extension scheme were comprehensively investigated by using a well-calibrated technology computer-aided design (TCAD).

## 2. Device Structure and Simulation Methodology

We used Synopsys’s Sentaurus to investigate the electrical and mechanical characteristics of the proposed scheme in sub-3 nm node NSFETs [22]. The following models were considered for accurate simulation:The drift-diffusion model was used to solve for the carrier concentration and electrostatic potential, coupled with the density-gradient model to reflect the quantum confinement effects [23,24].The Slotboom bandgap narrowing model was used because the bandgap narrows in highly doped silicon and silicon-germanium alloys [25,26].Mobility models were used to consider the scattering effects (mobility degradation at the interface, inversion, and accumulation layer mobility models) [27,28].The mobility models of electric fields (low-field ballistic mobility and high-field saturation models) were used to consider the quasi-ballistic transport and velocity saturation [29,30].Recombination models (Auger and Shockley–Read–Hall) were used to consider carrier generation and recombination.The deformation potential model was used because the band structure and effective mass of the electron/hole change according to the tensile/compressive stress [31].

The structures of NSFETs using the LSA and proposed S/D extension scheme are shown in Figure 1. Contact poly pitch (CPP) was set to 42 nm, and the fin pitch (FP) was set to 70 nm. NSFETs with buried oxides (BOX) were used to rule out the effects of the parasitic bottom transistor in the punch-through stopper (PTS) region [32,33], and the PTS region was doped at 5 × 10^18^ cm^−3^. The S/D doping concentration for the N- and PFETs was set to 4 × 10^20^ cm^−3^, and Si_0.98_C_0.02_ (Si_0.5_Ge_0.5_) was used to induce tensile (compressive) stress in NFETs (PFETs). The contact resistance was set to 1 nΩ/cm^2^, and the inner spacer thickness (T_IS_) was set to 5 nm [34]. The physical parameters were well calibrated on the basis of using advanced 5 nm node FinFETs, as shown in [35]. Physical parameters (ballistic coefficient and saturation velocity) that greatly affect the carrier transport were adjusted to calibrate the drain current. The annealing conditions were changed to match the electrical characteristics in the subthreshold region. The drain bias was set to |0.7| V (50 mV) for saturation (linear) operation. The geometric parameters used in the TCAD simulations are listed in Table 1.

The process flow of the proposed S/D extension scheme is shown in Figure 1b. After the dummy gate formation, sacrificial SiGe layers are partially etched to form the inner spacer. Next, the inner spacer is formed by filling the cavities with low-k material. As a result, the exposed surfaces of NS channels are filled by the low-k. Meanwhile, low-k deposited at the edge of the NS channels is etched during the S/D recess process (the structure (1) of Figure 1b). The feasibility of the proposed scheme can be improved by adopting low-k materials with high selectivity to silicon. Because a selectivity difference between the low-k and the silicon allows selective etching, only the NS channel can be etched by using the dry etch process (see the NS-channel-etching process in Figure 1b). Thereafter, S/D can be grown from the etched NS channels into the S/D region, resulting in extended S/D. Therefore, the junction between the S/D and the NS channels moves to the center of the NS with an increase in the etching depth of the NS channel (T_Si_E_), as shown in Figure 1a. Etching T_Si_E_ greater than the inner spacer thickness (T_IS_) can remove the NS channels as well as the SiGe sacrificial layers. Therefore, T_Si_E_ is split from 0 to 4 nm because the optimal T_IS_ for performance optimization is 5 nm [34]. Additionally, T_Si_E_ is split only for NSFETs with the LSA process because a slight S/D extension results in severe SCEs in the RTA process.

During the additional NS-channel-etching process, the proposed S/D extension scheme etches the exposed silicon regions of NS channels and PTS. Therefore, when the bottom oxide is not formed, an unintended S/D recess depth can increase the punch-through current [33,36]. However, these issues can be completely solved by forming the BOX. This is because BOX physically blocks an unintended leakage path through the PTS region. Therefore, it is desirable to use NSFETs with BOX in order to prevent an increase in punch-through current that is due to the proposed scheme.

We simulated two device structures assuming the following: (1) the RTA and (2) the LSA process conditions. First, NSFETs with RTA were simulated on the basis of our previous data [35], and the S/D annealing conditions were split into RTA and LSA. Because the S/D dopant gradient along the channel length direction is approximately 3–5 nm/dec in advanced devices [37], the RTA condition for the TCAD simulation was set to match the dopant gradient of advanced devices. S/D dopants were diffused under the same annealing conditions for both N- and PFETs. However, the phosphorus concentration gradient was considerably higher than that of boron, as shown in Figure 2a. This is because phosphorus has a slightly higher diffusivity than boron [38]. Additionally, because the S/D annealing process was performed before removing the SiGe sacrificial layer, Ge diffusion into the NS channel increased the phosphorus diffusivity. Second, the LSA process conditions were set on the basis of our LSA hardware data and applied to the TCAD process simulation. 

Figure 2b shows the phosphorus dopant concentrations as a function of the depth. The experiment was designed to investigate the dopant diffusions according to the activation process, and the dopant profile of phosphorus was measured by using secondary ion mass spectroscopy (SIMS). Phosphorus dopants were implanted into the wafer with a dose of 5 × 10^14^ cm^−2^ and an energy of 40 keV. The black line in Figure 2b represents the ion implantation results before the dopant activation processes, and the depth with a peak concentration of phosphorus ion (C_P_) was 60.7 nm. The blue line in Figure 2b shows the phosphorus profiles after the RTA thermal treatment. Phosphorus dopants were largely diffused, and the C_P_ was moved from 60.7 nm to 34.2 nm. Accordingly, because the RTA process can diffuse S/D dopants deeply into the channels, RTA should be avoided for advanced logic devices.

On the other hand, the C_P_ after LSA was 58.5 nm, and the difference from the phosphorus profile before thermal annealing was negligible. The diffusionless phosphorus profile can be achieved because the LSA activates the S/D dopants within a short dwell time. Therefore, a super-steep junction can be assumed for the TCAD simulation thanks to the diffusionless features. Therefore, a super-steep junction of less than 1 nm/dec was reflected, and fully activated S/D dopants were assumed for the TCAD simulation.

## 3. Results and Discussion

The transfer curves of NSFETs according to T_Si_E_ are shown in Figure 3. I_off_ was matched at 1 nA for a performance comparison. To simplify the notation, we defined NSFETs by using the RTA process as NS-RTA and NSFETs using the LSA process as NS-LSA. The drain current of the NS-RTA has a higher I_on_ than that of the NS-LSA when T_Si_E_ is lower than 2 nm. However, the I_on_ of NS-LSA significantly increased as T_Si_E_ increased; hence, NS-LSA with a T_Si_E_ of 2–4 nm outperformed NS-RTA. Additionally, although T_Si_E_ decreased the channel length, no significant SCEs were observed, owing to the diffusionless features of LSA. Therefore, the S/D extension scheme with the LSA can effectively improve the DC characteristics of NSFETs compared with the RTA.

As shown in Figure 4, I_on_ increases as T_Si_E_ increases, where I_on_ is the drain current value extracted at |V_gs_| = 0.7 V and |V_ds_| = 0.7 V. In NS-LSA at T_Si_E_ = 0 nm, I_on_ degraded by over 15% compared with NS-RTA. However, the I_on_ of NS-LSA could be improved by applying the proposed S/D extension scheme, and a considerably higher I_on_ was observed when T_Si_E_ was larger than 2 nm. I_on_ increased by 5.92% (17.1%) for the NFETs (PFETs) compared with the NS-RTA. Additionally, I_on_ increased by up to 21.7% (37.4%) in NFETs (PFETs) compared with NS-LSA (T_Si_E_ = 0 nm). Therefore, the benefit of using LSA with the S/D extension scheme makes it the best solution to boost the performance of NS-LSA. Notably, NS-LSA, which did not apply the proposed scheme (T_Si_E_ = 0 nm), exhibited a much lower I_on_ than NS-RTA.

The subthreshold swing (SS) according to T_Si_E_ and the conduction band energy (E_C_) of NFETs along the channel length direction to explain the reason for the I_on_ degradation in NS-LSA at T_Si_E_ = 0 nm are shown in Figure 5. The LSA minimized S/D dopant diffusion, and the SS was improved in NS-LSA compared with NS-RTA (Figure 5a). Because a high-energy barrier height (Φ_b_) is maintained thanks to the diffusionless S/D dopant in NS channels of NS-LSA, excellent gate controllability over the NS channels improved SS. Therefore, it is critical for advanced devices to suppress the S/D dopant diffusion because the lowered Φ_b_ results in SCEs and SS degradation. The SS was improved by 2 mV/dec (5 mV/dec) in the n-type (p-type) NS-LSA at T_Si_E_ = 0 nm compared with NS-RTA. However, SS significantly degraded as T_Si_E_ increased because the channel length was decreased during NS channel etching. Although the activation of S/D dopants was almost diffusionless, the shorter distance between the source and the drain induced degradation behavior. Consequently, the SS of NS-LSA exceeded that of NS-RTA when T_Si_E_ was greater than 2 nm, as shown in Figure 5a.

Although LSA provided better a SS value through its excellent gate controllability, I_on_ did not improve in NS-LSA (T_Si_E_ = 0 nm) compared with RTA (Figure 4). This is because the high- Φ_b_ owing to the lower dopant concentration in the S/D extension region resulted in a significant I_on_ reduction (Figure 5b). Φ_b_ was increased by 82 meV in NS-LSA; hence, Φ_b_ offset the improvement in SS. Therefore, a novel process scheme must be introduced to enable the use of the LSA. The reason for the increase in I_on_ for NS-LSA is shown in Figure 6 and Figure 7.

The increase in I_on_ with a large T_Si_E_ was induced by strain effects along the channel length direction (S_ZZ_). As the NS channels are etched, the S/D volume increases and the channel length decreases; thus, higher stress was induced in the NS channels (Figure 6a). The proposed S/D extension scheme effectively induced higher stress in the NS channels. The stress difference under the inner spacer region was small compared with that under RTA and LSA with T_IS_E_ = 0 nm; however, S_ZZ_ significantly increased when T_Si_E_ increased from 0 to 4 nm (Figure 6b). The average |S_ZZ_| at the center of the NS channels (top, middle, and bottom) is shown in Figure 6c; the S_ZZ_ values for the volume from −5 to 5 nm were averaged. As T_Si_E_ increased, an increase in stress of over 25% was induced for both N-/PFETs. The stress induced in the channel dominantly affects electron/hole mobility. In particular, the tensile stress along the channel length direction increases the electron mobility, and the compressive stress increases the hole mobility. Because the average tensile (compressive) stress over the NS channels was improved in NFETs (PFETs), higher stress can significantly increase electron (hole) mobility.

Carrier density was also a crucial factor for I_on_ enhancement. Figure 7 shows the electron density (eDensity) profiles according to the T_Si_E_ in NFETs, and eDensity profiles were extracted at an on-state bias (V_gs_ = V_ds_ = 0.7 V). In NS-LSA (T_Si_E_ = 0 nm), eDensity under the inner spacer was not large, because the S/D dopants were not largely diffused into the NS channels. On the other hand, eDensity under the inner spacer greatly increased as T_Si_E_ increased, and this is because a high-doped S/D region replaced the region with low-doped NS channel regions. This is greatly related to the extension region resistance because the parasitic resistance varies according to the extension region with low carrier concentrations. As a result, carrier concentrations in the NS channels of NS-LSA (T_Si_E_ = 4 nm) increased the I_on_ compared to NS-LSA (T_Si_E_ = 0 nm). Therefore, it can be concluded that the main factors increasing the I_on_ in the proposed S/D extension structure are the stress-boosting and carrier concentrations.

Although the L_g_ decreases according to the scaling of NSFETs, the T_IS_ cannot be freely scaled, owing to the trade-offs between the parasitic resistance and the parasitic capacitance [36]. Therefore, T_IS_ would be maintained at around 5 nm for sub-3 nm nodes and beyond, and the parasitic resistance could be the main factor hindering the scaling of the NSFETs. This is because the ratio of the extension region over the total NS channel region highly increases as the L_g_ scales down. As a result, controlling the parasitic resistance is the core technology for the further scaling of NSFETs. Because the proposed scheme controls the parasitic resistance by replacing the high-resistance regions with low-resistance regions, the proposed scheme can be used as a core technology for the further scaling of NSFETs.

Because the proposed scheme decreased the distance between the gate and the S/D, the electric field between the gate metal and the S/D significantly increased. The overlap capacitance (C_ov_) increased according to T_Si_E_, so the deeply formed S/D extension increased the gate capacitance (C_gg_), as shown in Figure 8a. Therefore, NS-LSA with T_Si_E_ = 0 nm can maximize the diffusionless advantage in terms of gate capacitance, but a decrease in the I_on_ offsets the reduction in the parasitic capacitance. The RC delay according to T_Si_E_ is shown in Figure 8b. In the NFETs, the RC delay significantly decreased as T_Si_E_ increased; however, a slight increase in the RC delay was observed at T_Si_E_ = 4 nm. This is because the SCEs and parasitic capacitance degraded the RC delay. However, in PFETs with T_Si_E_ = 4 nm, the RC delay can be improved by 9.2% compared with that using RTA. Compared with NS-RTA, C_gg_ was increased by 6.1% in NS-LSA (T_Si_E_ = 4 nm), but I_on_ was improved by 17.1%. As a result, RC delay degradation owing to the parasitic capacitance could be suppressed.

Although NS-LSA at T_Si_E_ = 0 nm exhibited the smallest C_gg_, RC delay cannot be improved, owing to the unintended high Φ_b_. A 24.2% RC delay degradation was observed when compared to RTA. This means that the LSA process can cause negative effects on device behaviors. Therefore, using the LSA process alone has no advantages in terms of AC/DC performances, but it is advantageous when used with a new process and device structures. The proposed S/D extension scheme, which adds only one process step, effectively decreased the RC delay by up to 9.27% through stress-boosting and high carrier concentrations in NS channels. Therefore, the S/D extension scheme is expected to be widely used to improve performance for further scaling.

## 4. Conclusions

This study proposed a novel S/D extension scheme using the LSA process for NSFETs, which increased I_on_ by boosting the channel stress and carrier concentrations. The LSA process activated S/D dopants without diffusion, so a new process scheme that was not feasible, owing to the S/D dopant diffusions, can be applied. NS-LSA (T_Si_E_ = 0) had an ultra-shallow S/D-NS junction, and the C_gg_ was improved thanks to the C_ov_ reduction. However, the I_on_ was degraded by 15.8% (20.3%) in NFETs (PFETs) owing to its high-energy barrier height. Although the SS was slightly degraded as the T_Si_E_ increased, a great increase in I_on_ was observed thanks to the higher stress and carrier concentrations compared with the NS-LSA (T_Si_E_ = 0 nm). On the other hand, the proposed S/D extension scheme shifted the junction toward the center of the NS channels. A large S/D volume induced higher stress and more carriers in NS channels compared to NS-RTA. Consequently, I_on_ increased by up to 5.92% (17.1%) in NFETs (PFETs) thanks to the strain effects, compared with NS-RTA. Furthermore, although the C_gg_ increased, owing to the increase in the overlap capacitance, as T_Si_E_ increased, a better enhancement in I_on_ improved the RC delay by 2.03% (9.27%) in NFETs (PFETs) compared with NS-RTA. Hence, the proposed S/D extension scheme is expected to be a key technology for improving the performance of NSFETs in heterogeneous 3D ICs.

## Figures and Tables

**Figure 1 nanomaterials-13-00868-f001:**
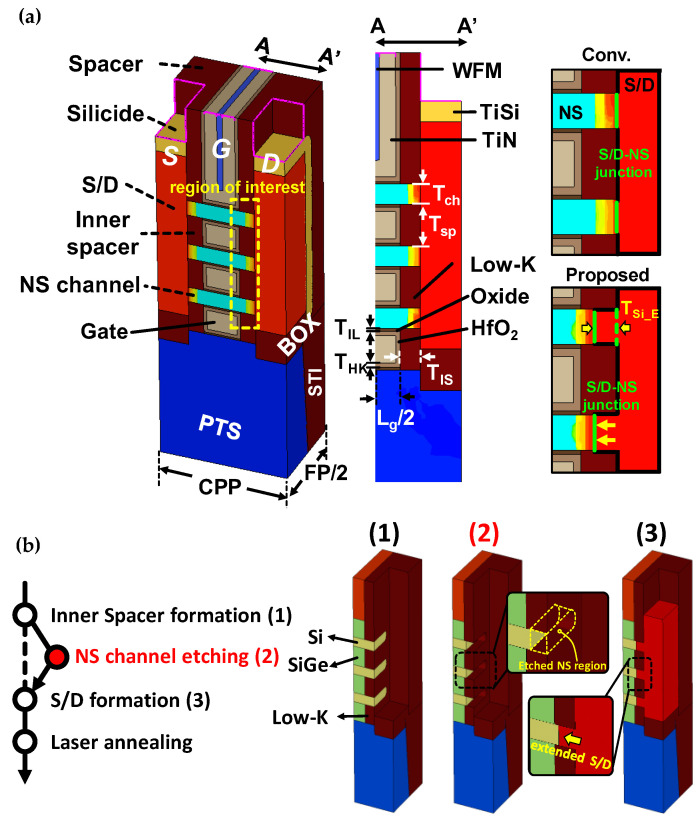
(**a**) Conventional and proposed structures of NSEFTs and their cross-sectional views. (**b**) The process flow of the S/D extension scheme. The added process step and etched NS region are highlighted in the red text and dashed yellow boxes.

**Figure 2 nanomaterials-13-00868-f002:**
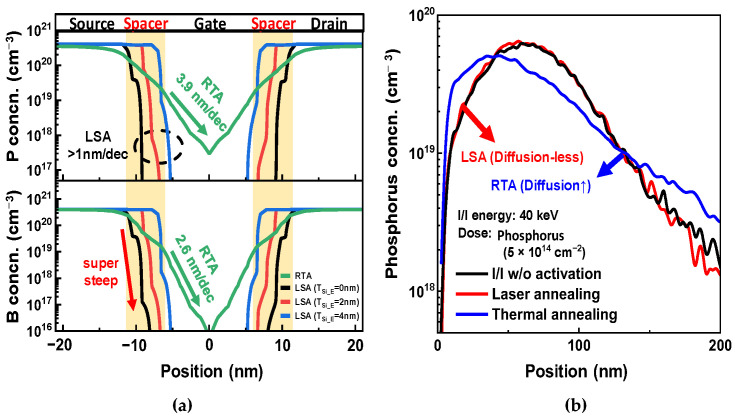
(**a**) Phosphorus and boron profiles along the source-drain direction after annealing. (**b**) SIMS profile of phosphorus for the calibration of the LSA and RTA processes.

**Figure 3 nanomaterials-13-00868-f003:**
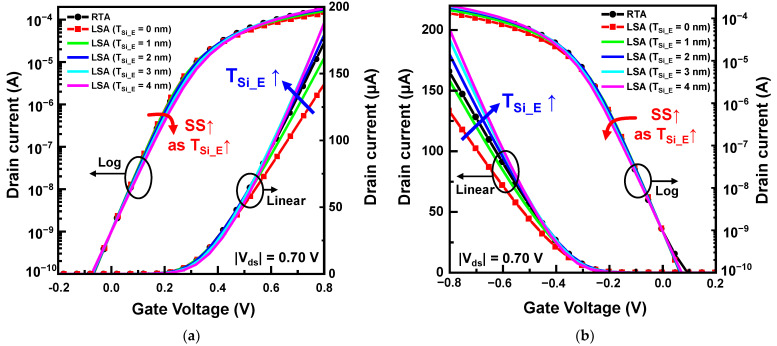
Transfer curves of NSFETs according to the T_Si_E_ for (**a**) NFETs and (**b**) PFETs.

**Figure 4 nanomaterials-13-00868-f004:**
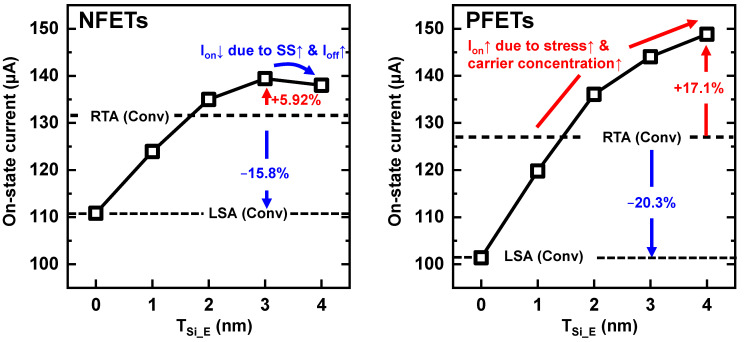
I_on_ according to T_Si_E_ for N-/PFETs. The dashed line denotes the I_on_ of NS-RTA and NS-LSA (T_Si_E_ = 0 nm), and the difference in I_on_ when using LSA versus RTA is indicated as a red/blue arrow.

**Figure 5 nanomaterials-13-00868-f005:**
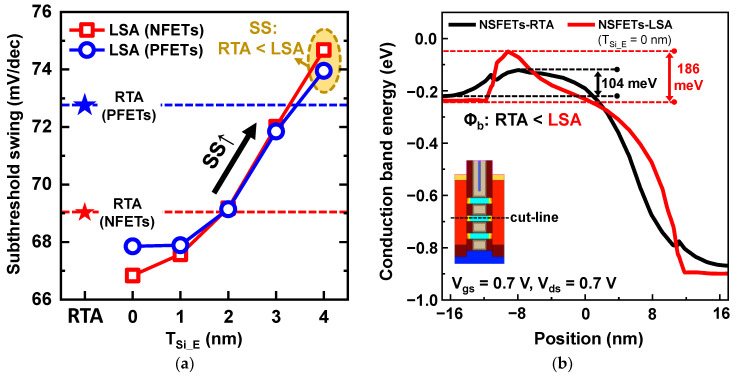
(**a**) SS values according to T_Si_E_ for N-/PFETs; the SS values of NS-RTA were indicated as star symbols. (**b**) Conduction band energy of n-type NS-RTA (black line) and NS-LSA (red line), and the conduction band energy was extracted at the center of the NS channels along the channel length direction.

**Figure 6 nanomaterials-13-00868-f006:**
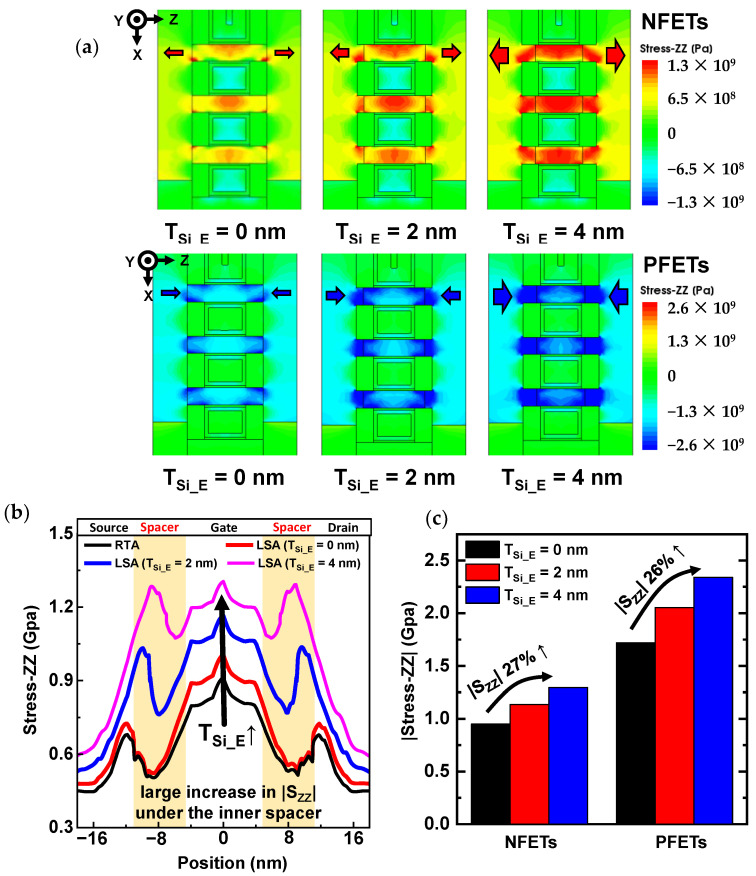
(**a**) Stress profile along the channel length direction (S_ZZ_) in NS-LSA. (**b**) Stress along the NS channel length direction in NFETs. (**c**) Averaged |S_ZZ_| value in NS channels below the gate metal.

**Figure 7 nanomaterials-13-00868-f007:**
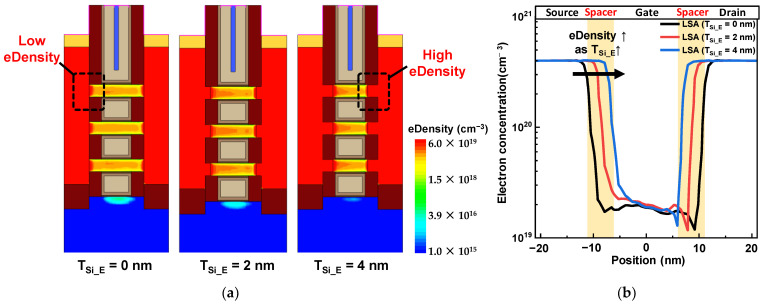
(**a**) Electron concentration in the NS channels according to the T_Si_E_ and (**b**) electron density profile along the channel length direction. Areas with a large difference in carrier concentrations are highlighted with dashed boxes.

**Figure 8 nanomaterials-13-00868-f008:**
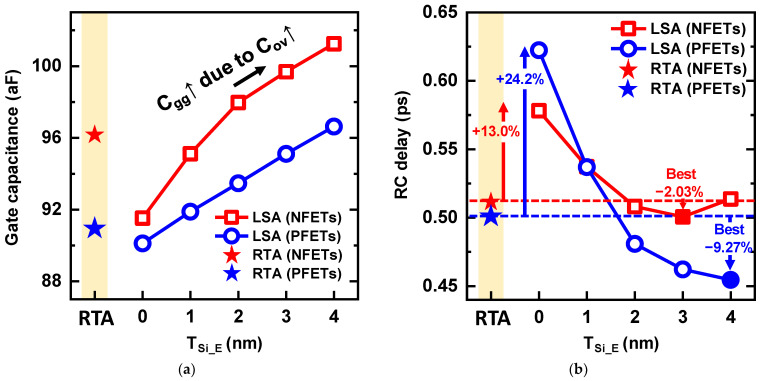
(**a**) Gate capacitance and (**b**) RC delay of NSFETs according to the T_Si_E_. The dashed line indicates the RC delay of NS-RTA, and the solid symbol indicates the optimum point showing the lowest RC delay.

**Table 1 nanomaterials-13-00868-t001:** Fixed and variable geometric parameters for sub-3 nm node NSFETs.

Fixed Parameters	Values
Contact poly pitch (CPP)	42 nm
Fin pitch (FP)	60 nm
Gate length (L_g_)	12 nm
Spacing thickness (T_sp_)	10 nm
NS thickness (T_ch_)	5 nm
Interfacial layer/HfO_2_ thickness (T_IL_/T_HK_)	0.6 nm/1.1 nm
Inner spacer thickness (T_IS_)	5 nm
S/D doping concentration (N_SD_)	4 × 10^20^ cm^−3^
PTS doping concentration (N_PTS_)	5 × 10^18^ cm^−3^
**Variable Parameters**	**Values**
Etch depth of the NS channel (T_Si_E_)	0–4 nm

## Data Availability

Not applicable.

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
