# Peer review of "A Novel Source/Drain Extension Scheme with Laser-Spike Annealing for Nanosheet Field-Effect Transistors in 3D ICs"

_nanomaterials, 2023, doi:10.3390/nano13050868_

Round 1
Reviewer 1 Report
The article "A Novel Source/Drain Extension Scheme with Laser Spike An-2 nealing for Nanosheet Field-Effect Transistors in 3D-ICs" by Lee et al. proposes a new source/drain (S/D) extension method to increase the stress in nanosheet (NS) field-effect transistors (NSFETs) to improve the transistor performance by enhancing the on-state current and reducing RC delay. The work is of great importance in the current semi-conductor industry as the results could potentially improve the performance of semiconductor devices from fundamental perspectives. The authors provided thorough introduction on simulation methods and discussion on the results. The conclusion is persuasive based on the results given. In general, this is a high quality paper and I would like the authors to address a few more points for improvement.
1. All analysis are based on simulation results, which is understandable as this field is tough to access through experimental perspective. However, I do think it would be nice for authors to discuss the possibility of achieving this in real semiconductor industry with potential engineering barriers to be solved. In this way, this paper may have more impact in future.
2. Line 39-40 "A higher performance" is very vague, please be more specific on what aspect of performance is better.
3. Line 221-222. The sentence is not clear on discussion, wasn't sure the meaning.
Reviewer 2 Report
In this paper, the authors propose an original source/drain extension scheme to overcome the Ion reduction issues in three-dimensional integrated circuits by increasing the stress and the carrier concentration in nanosheet field-effect transistors. They demonstrate the concept through computer-aided design simulations.
The subject is interesting but best suited to an electronic journal. The paper is clear and the study is well conducted. The paper can be published after a minor revision.
“After the inner spacer was formed, an additional NS channel etching process was performed and 100 S/D materials were filled from the etched NS to the S/D region.” This is the key point of this proposal. The authors should give more details. How could this additional NS channel etching be performed in reality?
“Because the LSA minimized S/D dopant diffusion, the SS was improved in NS-LSA compared with NS-RTA (Fig. 5a)” How is the SS related to S/D dopant diffusion? Please clarify
“As the NS channels are etched, the S/D volume increases and the channel length decreases; thus, larger stress was induced in NS channels (Fig. 6a).” This is clear. However, the authors could emphasize that tensile and compressive stress is considered for NMOS and PMOS respectively.
